# Real-time counting of wheezing events from lung sounds using deep learning algorithms: Implications for disease prediction and early intervention

Sunghoon Im[1], Taewi Kim[1], Choongki Min[2], Sanghun Kang[1], Yeonwook Roh[1], Changhwan Kim[1], Minho Kim[1], Seung Hyun Kim[3], KyungMin Shim[4], Je-sung Koh[1], Seungyong Han[1], JaeWang Lee[5], Dohyeong Kim[6]*, Daeshik Kang[1]*, SungChul Seo[7]*

1 Department of Mechanical Engineering, Ajou University, Suwon-si, Gyeonggi-do, Republic of Korea, 2 Waycen, Inc., Seoul, Republic of Korea, 3 Department of Medical Humanities, Korea University College of Medicine, Seoul, Republic of Korea, 4 Industry-University Cooperation Foundation, Seogyeong University, Seoul, Republic of Korea, 5 Department of Biomedical Laboratory Science, College of Health Science, Eulji University, Seongnam-si, Gyeonggi-do, Republic of Korea, 6 University of Texas at Dallas, Richardson, TX, United States of America, 7 Department of Nano-Chemical, Biological and Environmental Engineering, Seogyeong University, Seoul, Republic of Korea

☯ These authors contributed equally to this work.
* dohyeong.kim@utdallas.edu (DK); dskang@ajou.ac.kr (DK); haha0694@gmail.com (SS)

**Data Availability Statement:** Codes and utilized data are available in our open repository (https://github.com/sunghoon-most/Wheeze_Counter).

## Abstract

This pioneering study aims to revolutionize self-symptom management and telemedicine-based remote monitoring through the development of a real-time wheeze counting algorithm. Leveraging a novel approach that includes the detailed labeling of one breathing cycle into three types: break, normal, and wheeze, this study not only identifies abnormal sounds within each breath but also captures comprehensive data on their location, duration, and relationships within entire respiratory cycles, including atypical patterns. This innovative strategy is based on a combination of a one-dimensional convolutional neural network (1D-CNN) and a long short-term memory (LSTM) network model, enabling real-time analysis of respiratory sounds. Notably, it stands out for its capacity to handle continuous data, distinguishing it from conventional lung sound classification algorithms. The study utilizes a substantial dataset consisting of 535 respiration cycles from diverse sources, including the Child Sim Lung Sound Simulator, the EMTprep Open-Source Database, Clinical Patient Records, and the ICBHI 2017 Challenge Database. Achieving a classification accuracy of 90%, the exceptional result metrics encompass the identification of each breath cycle and simultaneous detection of the abnormal sound, enabling the real-time wheeze counting of all respirations. This innovative wheeze counter holds the promise of revolutionizing research on predicting lung diseases based on long-term breathing patterns and offers applicability in clinical and non-clinical settings for on-the-go detection and remote intervention of exacerbated respiratory symptoms.

**Funding:** This work was supported by the Korea Environment Industry & Technology Institute (KEITI) through the Environmental Health Digital Program funded by the Korea Ministry of Environment (MOE) (2021003330010, 2021003330009) The funders had no role in study design, data collection and analysis, decision to publish, or preparation of the manuscript.

**Competing interests:** The authors have declared that no competing interests exist.

## Introduction

Lung diseases are a major cause of global morbidity and mortality, including asthma, COPD, lung infections like pneumonia, lung cancer, bronchitis, and other breathing problems [1, 2]. Lung sounds can be indicative of most lung and respiratory diseases [3]. When there is no respiratory disorder, normal breathing sounds are heard, whereas abnormal breathing sounds such as wheezing or crackling are detected when there is a lung disease [4, 5]. For this reason, regular or routine monitoring of breathing sounds is essential for symptom prevention and alleviation, as well as for the early detection of various respiratory diseases [6, 7]. Typically, respiratory abnormalities are diagnosed by spirometry and auscultation [8]. While spirometry is impossible for certain groups, such as children, and is difficult to use practically to monitor a long-term pattern of patient condition in non-clinical settings [9, 10], auscultation is non-invasive, inexpensive, and easy to use [11, 12]. Medical professionals listen to these sounds to evaluate and diagnose patients [13]; however, conventional auscultation requires considerable training and expertise, and its quality depends on the doctor's experience and hearing [14]. The misunderstanding of breathing sounds and making incorrect diagnoses is not rare among medical students [15, 16].

To overcome the limitation of conventional auscultation, various methods such as neural networks [17], classifiers [18, 19], and NMF [20] are suggested in many cases in order to assist in the automatic detection and classification of adventitious lung sounds [21]. Among them, deep learning algorithms train the machine to automatically learn the characteristics of the signals or waveforms of lung sounds to recognize abnormal lung or breathing sounds (wheezing, crackling) [22]. The most common deep learning algorithm used for lung sound classification is a convolutional neural network (CNN) [11, 15, 23, 24] or recurrent neural network (RNN) model [25, 26] that extracts breathing sound features from a two-dimensional spectrogram image, or a combination of the two, a convolutional-recurrent neural network (CRNN) [22, 27]. The accuracy of the models ranges from 63% [11] to 99% [28], and in general, the CNN-based model has the highest accuracy [5]. Incorporating AI-based lung sound analysis into automated diagnosis systems has been suggested to determine the degree of airway inflammation [29] or the risk of a number of lung diseases [30]. Recently, efforts have been made to collect breathing sounds from smartphones or real-time lung sounds from wearable devices to develop automated AI-based solutions for lung sound analysis and classification [31–34]. Through this technological advancement, abnormal respiratory and asthmatic symptoms could be detected or diagnosed at an early stage via real-time self-monitoring or telemedicine [35, 36].

However, most existing models focus on the automatic diagnosis of single recorded data, and applications to real-time monitoring data are still limited [21, 37]. They tended to be developed based on the learning data collected by auscultation for a short period of 10 to 70 s and labeled by clinicians [38, 39]. Much of the previous work focused on addressing methodological challenges associated with noise cancellation or reduction [40, 41], detection of the breathing section, or binary classification of an individual cycle of respiration [11, 22, 23, 42, 43]. Due to a lack of adaptability for real-time, continuous long-term signals, most lung sound classification algorithms have not been widely implemented in practice, with limited applicability in self-symptom management or telemedicine [2, 44]. Considering that respiratory patterns represent the holistic physical and psychological state of humans, not only the presence of abnormal sounds but also the location, duration, and relationships of a sequence of respiration cycles, including atypical breathing activities, could serve as important reference data for clinicians and patients to diagnose and monitor lung diseases [45]. To provide comprehensive information about the lung's breathing functionality, which may not be well noticed or

recognized in a clinical setting, the pattern and frequency of abnormal lung sounds within a relatively long time must be analyzed rather than most of the existing models for determining the presence or absence of abnormalities at each respiratory unit [46, 47]. The real-time data collection and automated pre-processing system would be critical for long-term monitoring and intervention [48]. We have summarized the relevant papers in a table and included them in the S1 Table.

Considering this loophole, in this exploratory study, we have developed a real-time event counting algorithm to identify abnormal breathing sounds, especially wheezing, and record their frequency to determine the pattern over a certain period and present this information in real-time. We utilize a unique method that involves the meticulous categorization of a single breathing cycle into three types: break, normal, and wheeze. The algorithm not only detects abnormal sounds in each breath but also collects extensive data on their location, duration, and connections within the entire respiratory cycle, including unusual patterns. This counting algorithm may improve existing studies that aim to predict lung diseases based on long-term breathing patterns [49–51], going beyond simply classifying respiratory units. In addition, when integrated with wearable devices that are being actively developed, its utility will be maximized [52, 53]. Using three types of labeled lung sound data, we trained a one-dimensional convolutional neural network and a long short-term memory (1D-CNN-LSTM) network model for discriminating three breathing statuses (break, normal, and wheezing) and then developed a "real-time wheezing counter" as a pilot; we suggested the possibility of its application for early diagnosis or the remote treatment of respiratory diseases. Our research demonstrates the potential of AI-based technology for diagnosing and monitoring lung diseases in real-time, offering the prospect of earlier detection and improved treatment outcomes. Existing research gaps include limitations in real-time applications and a focus on short-term data. We address these gaps with a real-time event counting algorithm designed for continuous, long-term signals, emphasizing the pattern and frequency of abnormal lung sounds over time, rather than just detecting their presence or absence at individual respiratory units. This advancement holds promise for enhancing the diagnosis and monitoring of lung diseases.

## Methods

### Overview

The procedure of the developed wheeze counting algorithm is illustrated in Fig 1. We first obtained multiple reference lung sound data sets from open sources and clinical data. We augmented the data using the pitch shift method to overcome the limited quantity of training samples. We then extracted the features of the augmented lung sound data using a Mel frequency spectrogram, which is widely used in sound analysis [54]. The preprocessed data were fed into a combined model of 1D-CNN-LSTM, which has been shown to be effective for lung disease recognition [55]. After sufficient training to ensure reliable accuracy using validation datasets, we tested the trained model with the test dataset. Finally, we developed and improved a wheeze counting algorithm that analyzes lung sound data to count the number of wheezes from clinical lung sound data. The algorithm could be applied to the long-term monitoring of breathing functions in clinical and non-clinical settings.

### Clinical lung sound data in this study

We employed a subset of the clinical lung sound data collected on November 30, 2021. for both training and testing purposes in our study. The first time we accessed the data was on March 10, 2022. To ensure the privacy and confidentiality of the participants, none of the authors had access to any information that could potentially reveal their identity. The

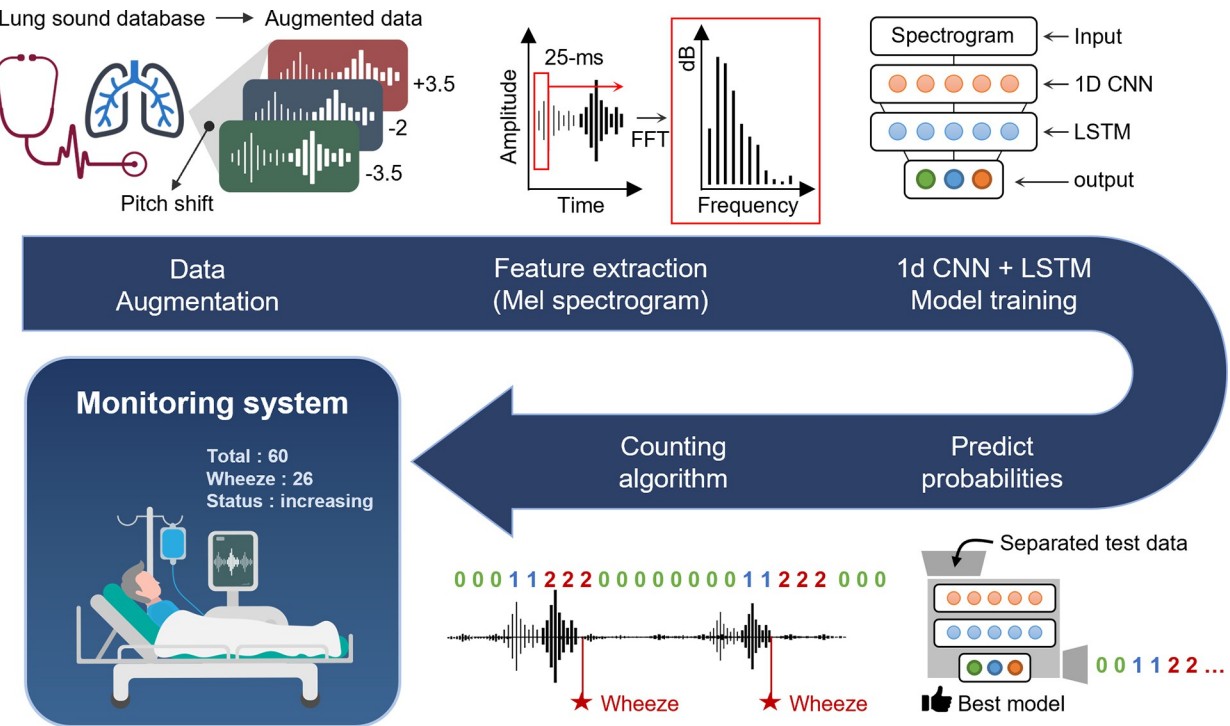

**Fig 1. Overall procedure of wheeze counting algorithm development and applications.**

Institutional Review Board at Eulji University approved the study, affirming that it was conducted in compliance with all relevant ethical standards.

## Lung sound databases

We obtained reference lung sound signals from three databases: 1) lung sound simulator, 2) EMTprep, 3) clinical patient records, and 4) ICBHI 2017 datasets. First, using a commercial microphone, ten and seven cycles of typical breathing and wheezing data were taken from the pediatric lung sound simulator known as Child Sim (SimulAIDS Inc, UK). Second, the open-source lung sound database EMTprep (https://www.EMTprep.com) provided three cycles of typical breathing and nine cycles of wheezing sound data. In the case of clinical data, we used 17 cycles of wheeze breathing using a commercial microphone that was affixed to the anterior right lung region. And last, we utilized additional diagnostic data from the ICBHI 2017 challenge database [31]. In the database, we used cases of asthma, COPD, and healthy patients. The various lung sound signals that were employed in this study are listed in Table 1.

## Soft labeling and data augmentation

Since some lung sounds annotated as "expiratory wheezing" contain both normal sound and adventitious sound in the isolated breathing cycle (S1 Fig), we soft-labeled the data manually

**Table 1. Number of lung sound signals by database.**

| Type | Simulator | EMTprep | Clinical | ICBHI | Total |
|---|---|---|---|---|---|
| Normal respiration | 10 | 3 | – | 287 | 300 |
| Wheeze respiration | 7 | 9 | 17 | 202 | 235 |

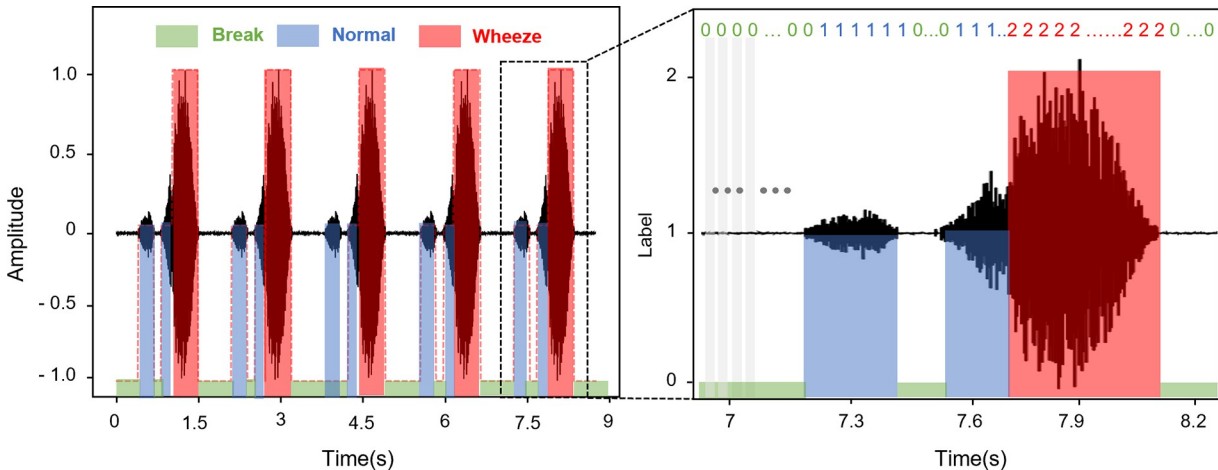

**Fig 2. Schematics of soft-labeling, and hard-labeling process.** The left graph shows 9 seconds of example lung sound and soft labeled annotation. The right graph shows the magnified scope (from 7 to 8.2 seconds) of the left graph with a hard labeled annotation.

before augmenting it. Based on the clinician's diagnosis, we used the free software Audacity to index the shifting boundary between normal and wheeze breathing during one breath cycle. We designated them specifically as "normal," "wheeze," and "break," which are depicted in blue, red, and green, respectively, in the left graph of Fig 2. Then, we augmented the soft labeled data using the Librosa Python library's "pitch shifting" function to get around data constraints [56]. In accordance with an earlier study, we changed the pitch by four semitone values (-3.5, -2, 2, and 3.5) [57, 58].

## Hard labeling and feature extraction

For each augmented lung sound, we sliced it into 25-ms segments with 10-ms overlaps. This allowed us to determine one breath cycle and, in addition, to determine which part of the breath contains accidental sound. Then we hard-labeled the pre-processed lung sound data based on soft labels. As illustrated in the right graph in Fig 2, all segments were hard labeled to 0, 1, or 2. The label ratio (the ratio of hard labeled segments among the soft labeled range) is chosen at 1 for high accuracy (S2 Fig). The segments in the 'break' range were labeled to 0, and in the same manner, the segments in the 'normal' range and 'wheeze' range were labeled into 1 and 2 each. Table 2 shows the total number of hard-labeled segments after hard labeling. From 535 breathing cycles in four databases, 332,720 segments were prepared as a machine learning dataset. Finally, we converted the segments into Mel spectrograms in order to extract acoustic features. 128 Mel bands were employed. Based on the effectiveness of the results from the counting algorithm, we select the proper parameter values of segment length and the number of Mel frequency bins (S2 Fig) for feature extraction.

**Table 2. Number of hard-labeled segments by database.**

| Label | Simulator | EMTprep | Clinical | ICBHI | Total |
|---|---|---|---|---|---|
| **Break (0)** | 899 | 399 | – | 253,998 | 255,296 |
| **Normal (1)** | 699 | 708 | – | 43,089 | 44,496 |
| **Wheeze (2)** | 159 | 1144 | 1844 | 29,781 | 32,928 |

## Results

### Model structure

We used a combination 1D-CNN-LSTM model including two blocks: CNN and LSTM blocks. The CNN block consists of 1d CNN layers, MaxPooling, and a dropout layer, while the LSTM block, connected directly to the CNN block, consists of LSTM, fully connected, and dropout layer. The MaxPooling and dropout layers were used to prevent overfitting. We used the Soft-Max function [59] as an activation function for the final output layer, finally yielding 3 dimensional outputs. Fig 3 shows the entire model structure. We did not shuffle the dataset during the training procedure, so the association over time could be sufficiently reflected. As a result, 25-ms segmented lung sounds were fed into the model, and the model predicted the probability of each class label. To create the model architecture, we employed the TensorFlow framework [60]. S2 Table lists the parameters of the neural network model, including the filter, kernel, and layer-specific unit sizes. We used a kernel size of 16 (about 1/10 of the input length) and 32 sets of filters, which is a regularly employed number, considering the size of the input data. The ReLu function is used as the activation function for each 1D CNN layer. We extracted the dominating weights through the MaxPooling layer, followed by 256 units of the bi-directional LSTM layer, after passing two layers of the 1D CNN. Then, ReLu was applied to 128 dense layer units for the following fully connected layers.

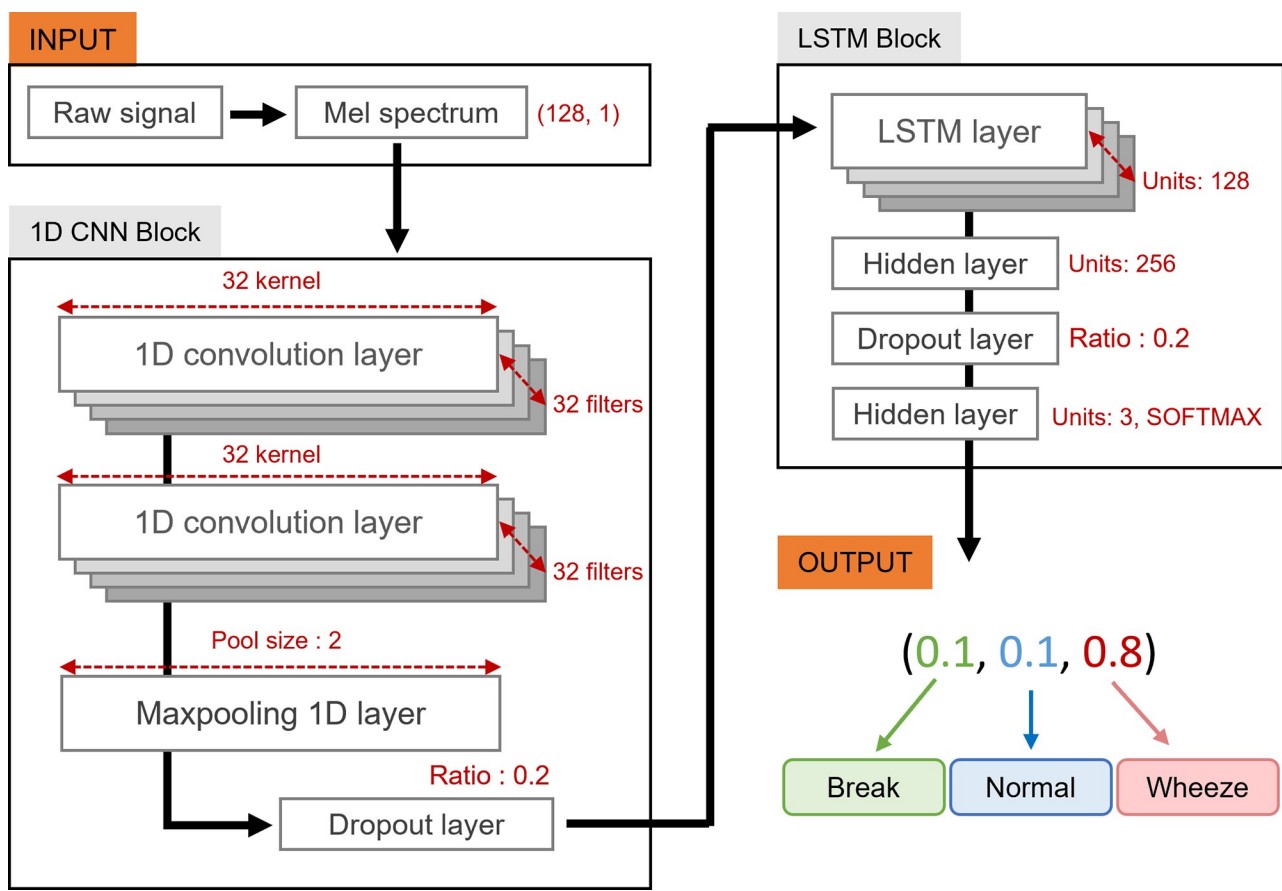

**Fig 3. 1D CNN and LSTM modeling structure and procedure.**

## Model training and validation

The hard-labeled dataset was divided into training and validation data. We used the 10-fold cross-validation method. The k-fold cross-validation method is beneficial as it ensures every data point is tested at least once and minimizes bias, providing a more accurate measure of the model's performance. This method also helps mitigate overfitting by using most of the data for fitting and testing each data point at least once. We utilized the model with the highest accuracy among the trained models. Then we evaluated the trained model's performance using test data by calculating its accuracy score, F1 score, and ROC-AUC (Area Under Receiver Operating Characteristic Curve) score. The test data is extracted from reference audio signals, which are totally unseen during the training process (Fig 5A). The F1 score is widely used as a performance measure in multi-label classification problems [61], and the ROC-AUC score is generally used for evaluating the performance of multi-label classifiers [62], that should be calculated using either "one-vs-rest" or "one-vs-one" methods [63]. In this study, we used it as "one-vs-rest". The prediction results of the trained model are shown in Fig 4. The model's

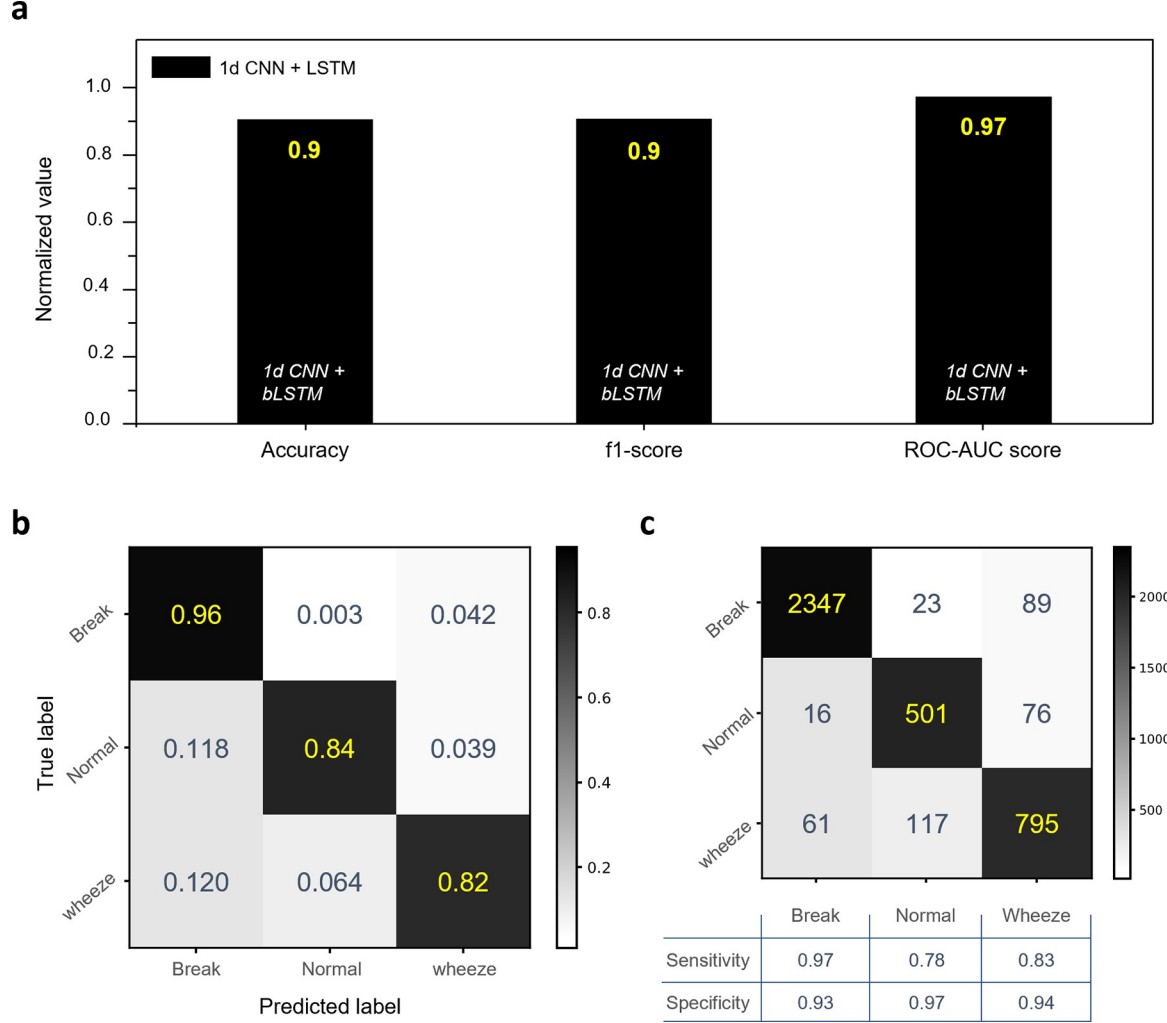

**Fig 4. Performance of the trained 1D CNN+LSTM model.** (A) Evaluation in three indicators: Accuracy, F1-score, and ROC-AUC score. (B) Confusion matrix of test data with normalization, and (C) without normalization and calculated sensitivity and specificity of each label.

accuracy score was 0.9, the F1 score was 0.91, and the ROC-AUC score was 0.98 (Fig 4A). We made use of the Sci-kit Learn Metrics Libraries to determine each score. The model's accuracy for each label is displayed via the model's normalized confusion matrix (Fig 4B). The model's accuracy for "wheeze" labels was 0.82, and for "normal" and "break" labels, it was 0.84 and 0.96. Each label's sensitivity and specificity were also determined using a confusion matrix without normalization (Fig 4C). The "wheeze" label has a sensitivity of 0.83 and a specificity of 0.94 for the test datasets. At the label of "normal," the specificity was at its highest, 0.97. Additionally, prior to applying the trained model to the counting algorithm, we also compared it to three popular and verified classifiers from the Sci-kit Learn library: Random Forest Classifier, K-Nearest Neighbors, and Multi-layer Perceptron classifier (S3 and S4 Figs). To further accurately assess the performance of those models, we used identical test datasets for each. In every indicator, the results demonstrated that the 1dCNN + LSTM model performed better than the others. Because of this, we used the model to implement our algorithm. Furthermore, in order to evaluate its durability in noisy environments, we also overlapped clinical environment noise from free open source (https://freesound.org/people/bassboybg/sounds/201638/, S5 Fig). to the test data. As shown in S6 Fig, the model misidentified the noise sound as wheezing, making it difficult to count the number of wheezings. However, after noisereduce-Python-library pre-processing, the model closely identified the true labels of the test data with less error. These findings show possible effectiveness in noisy environments when the model is combined with a proper noise cancellation method.

## Validation through visualizing predicted probabilities

The trained model successfully predicted each class label ("break," "normal," and "wheeze"), as shown in Fig 5. For the test, 10-second-long clinical lung sound recordings within four cycles of breaths were used (Fig 5A). The first breath (0 to 1.2 seconds) is normal, while the following breaths are abnormal, as can be seen in the Mel spectrogram image below the raw signal (Fig 5B). Then the input was segmented into 25-ms segments with 10-ms of overlap, and each segment was pre-processed for feature extraction by the same method as described in the preceding sections. Predict-to-break probabilities are shown as green, Predict-to-normal probabilities are shown as blue, and Predict-to-wheeze probabilities are shown as red in Fig 5C, while the reference probabilities for each label are shown as a dotted line. The findings demonstrate that the 1D-CNN-LSTM model's predicted probability accurately tracked the references' paths.

## Counting algorithm

We counted the frequencies of wheeze throughout the full lung sound recording using the predicted probabilities from the trained model. A ($N$, $128$, $1$) tensor encoding the recorded lung sound signals was generated, and it was then fed into the trained model. The trained model predicted probabilities in the shape of ($N$, $3$), with N being the number of lung sound segments after receiving the pre-processed input data. Our proposed counting algorithm is described as pseudo-code in Algorithm 1. Following are the steps: the segments were predicted in a ($N$, $3$) shape of outputs, and by using the argmax function in the NumPy library [64], the results were converted into a ($N$, $1$) shape of highest predicted labels (*0 or 1 or 2*); then, the peaks from the highest predicted labels were found by using the 'find peaks' function in the SciPy Library [65]; one breath cycle was considered complete when the value of d (peak interval) was longer than 100 (about one second); finally, isolated breathing was classified as "wheeze" if the average peak heights from separated breathing's range is higher than 1. If the average of peak is equal to 1, we considered the breath to be "normal."; In this manner, the total number of breath

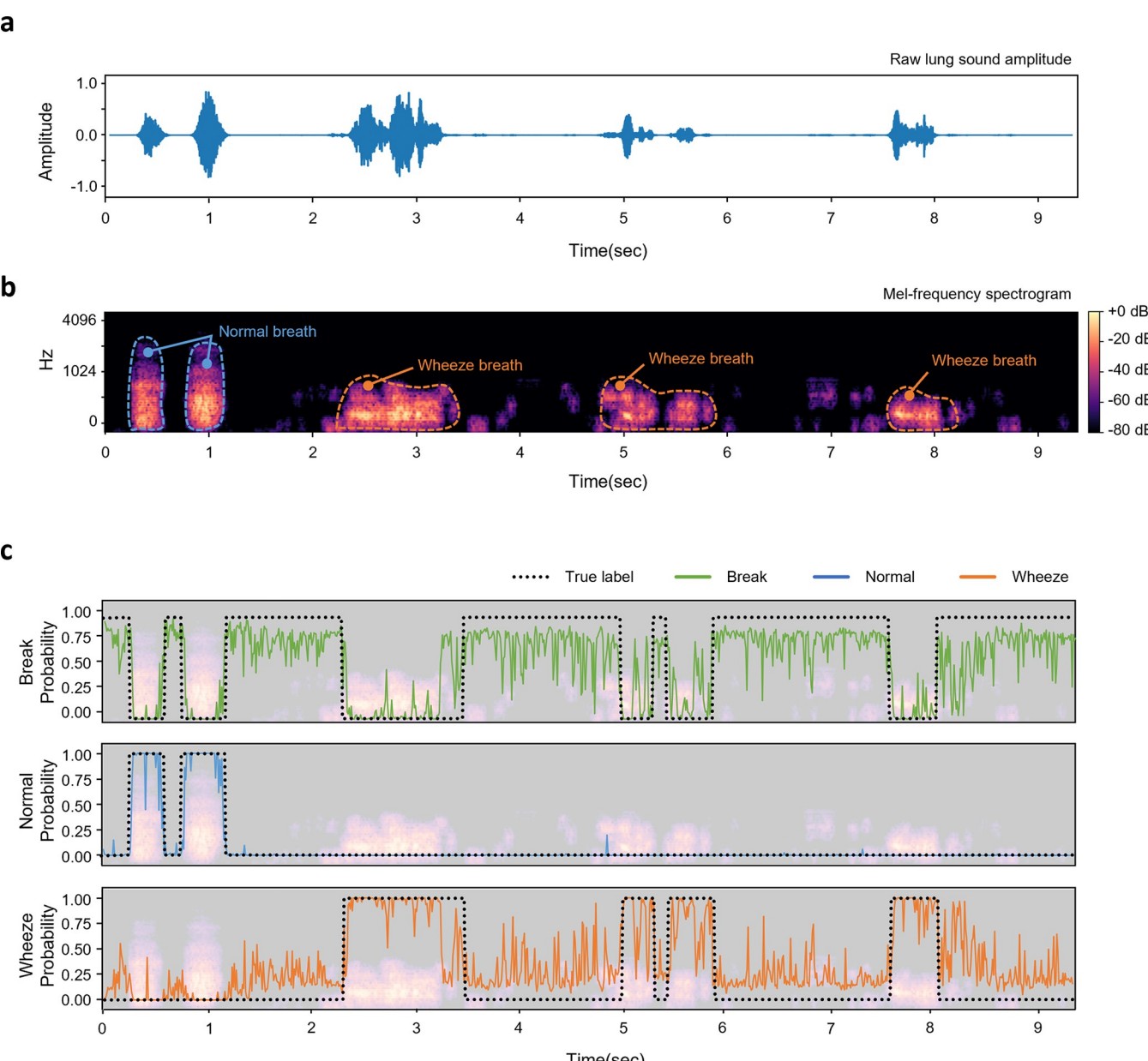

**Fig 5. Predicted probabilities of the 1d CNN+LSTM model.** (a) Raw data from the input and (b) Mel spectrogram. (c) Predicted probabilities from the trained 1D-CNN-LSTM model.

cycles as well as the total number of normal and wheeze events throughout the full record were captured.

```
Algorithm 1. Wheeze counting algorithm for the recorded lung sound.
def wheeze counter (calculated probabilities = (N,3))
        (N, 1) ← argmax (N, 3)
        d = peak interval
        if d ≥ 100
              count +1 respiration
              avg_p ← average value of peaks
```

```
if avg_p > 1.0
        count +1 wheeze
else if avg_p = 1.0
        count +1 normal
```

Additionally, a second algorithm for real-time counting to track wheeze events was created. We defined a Boolean variable named "Wheeze toggle" that is initially set to "False" at the beginning. The real-time wheeze counting procedure is described as pseudo-code in Algorithm 2 and illustrated in Fig 6: The algorithm takes raw signals with a 0.5-second duration as input, and the "Wheeze toggle" maintained its state even as the input's value changed over time in order to link the prior signals with the current signal (Fig 6A). The raw input signal is converted into the Mel spectrogram, then the scale is changed to dBs before being fed into the trained model; The trained model predicts the probabilities of the 0.25-ms segments as a (*N, 3*) shape of output from 0.5-s long raw input signals. The first index value of the output denotes the probability that the 25-ms segment would "break," the second index value

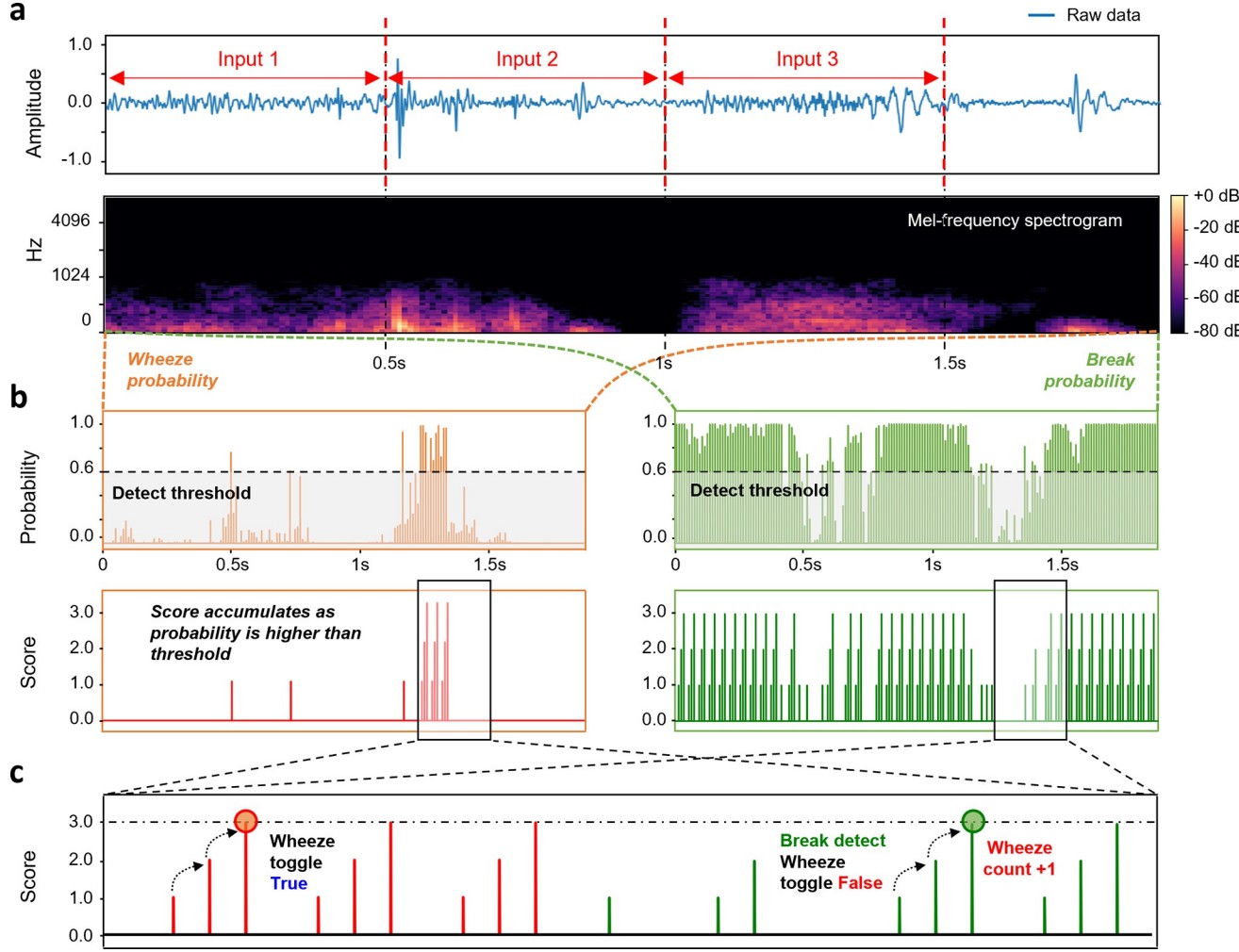

**Fig 6. Illustration of the real-time wheeze counting process.** (A) Raw signal of the clinical lung sound and Mel spectrogram. 0.5 seconds of input data is given to the model without overlap. (B) Score accumulates as probabilities are higher than threshold. The score resets to 0 after it reaches 3 points. (C) When the "Wheeze score" reaches 3 points, the "Wheeze toggle" changes to "True," and when the "break score" reaches 3 points, it turns back to "False".

denotes the probability that it will "normal," and the third index value denotes the probability that it will "wheeze." The total of the three probabilities is 1 since SoftMax is the activation function of the final fully connected layer. From the sequentially predicted probabilities, if predict-to-wheeze probability is highest 3 times in a row (Fig 6B), the moment is recognized as a wheeze occurrence and "Wheeze toggle" changes to "True." The breath cycle is then recognized to terminate, if probability of predict-to-break is also highest 3 times in a row, while the "Wheeze toggle" is "True." Then we reset "Wheeze toggle" to "False" for consecutive wheeze counting (Fig 6C). For real-time application, 0.5-s long lung sounds are sequentially fed into Algorithm 2 for wheeze occurrences to be continuously counted and presented to users.

```
Algorithm 2. Wheeze counting algorithm for real-time lung sound.
def real-time wheeze counter (raw audio signal)
  pred ← model (converted input)
      (break prob, normal prob, wheeze prob) ← pred
if wheeze prob ≥ 0.6 more than 3 times
                wheeze toggle 'True'
  while wheeze toggle 'False'
                if break pred ≥ 0.6 more than 3 times
                    wheeze toggle 'Off'
                    count +1 wheeze
```

## Application to clinical data

We monitored test data of relatively long-term clinical lung sounds and counted the number of wheeze occurrences using the real-time wheeze counting method established in this work (Fig 7A). The trained model was continuously fed lung sounds with 0.5-second durations, and

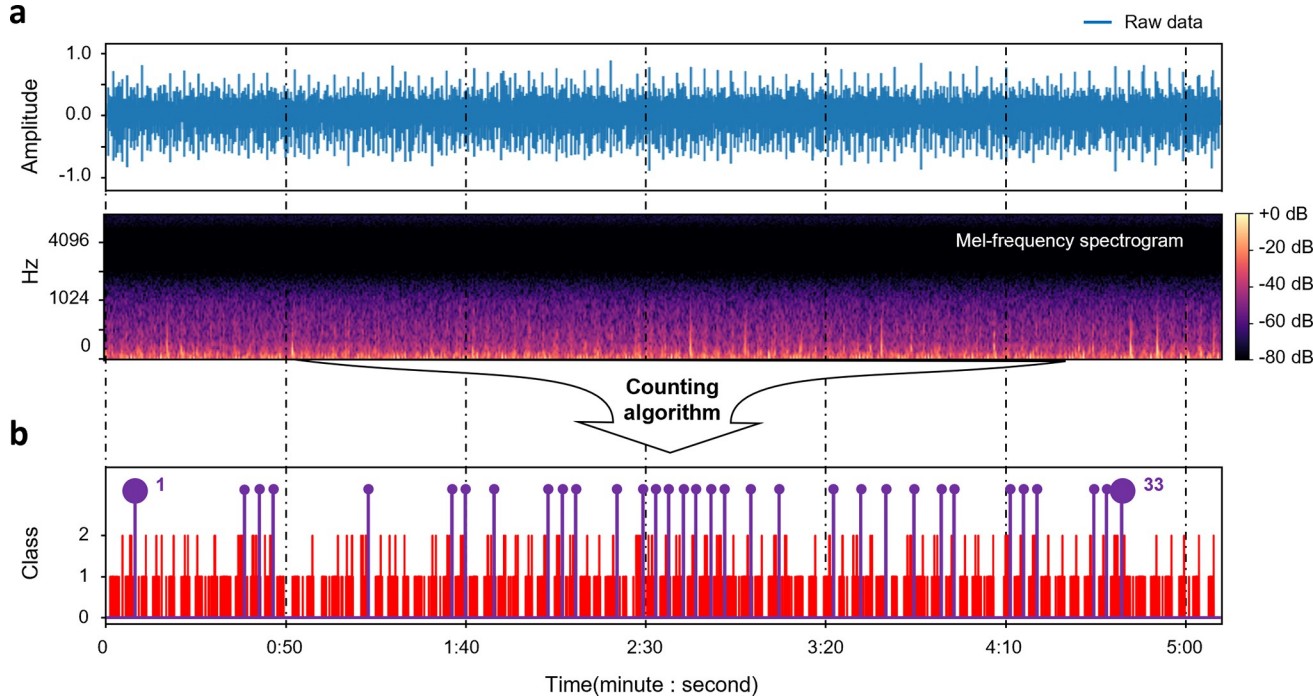

**Fig 7. Illustration of real-time wheeze counting using long-term clinical data.** (A) Raw signals of lung sound and Mel spectrogram for visualizing the acoustic features. (B) The results of our counting algorithm. The frequency of wheeze symptoms is counted, and time stamped.

using the counting algorithm, wheeze events were calculated and logged across the dataset. For 0.5 seconds of input, lengths of 8,000 samples were used to reshape the (51, 128, 1) shape of the input because the raw signal's sample rate was 16,000 Hz. Therefore, a (5, 128, 1) shape of input is fed into the model at intervals of 0.5 seconds, and the model predicts 51 sets of probabilities as outputs. As a result, the counting algorithm detected 33 wheezing occurrences from entire clinical lung sounds. From the 77 breaths detected by the algorithm, the wheeze rate was calculated at 0.43 (= 33/77) and was close to the wheeze rate noted by the doctor at 0.41 (= 32/77), that has only 2% of an error rate. Utilizing the results, our counting algorithm is simulated in real-time autonomous detection, as seen in the S1 Video.

## Discussion

Long-term monitoring of lung sounds assembled via a wearable device and AI-based diagnosis without doctor involvement would be essential to developing advanced computerized monitoring that may be used for self-symptom management or remote monitoring such as telemedicine. There have been a few research studies on these issues recently [16, 53]. However, the methods typically classify whether a signal is abnormal or normal or what kind of inadvertent sound it is. Their suggested method just distinguishes the subject's artificially induced inadvertent lung sound as a real-life applicable demonstration, indicating a practical usage limitation. Otherwise, we present an applicable method for implementing long-term monitoring in clinical settings by counting the number of wheeze occurrences over time. Our method differs from previous research in that it counts both the number of normal breaths and the number of wheezes, that is helpful for monitoring respiratory disease patients in dynamic environments. We utilize a segment-based classification AI model, which is normally used in speech recognition [66] or rare sound detection [67, 68]. To be able to detect not only wheezing events but also the isolated breath cycle, we sliced lung sound signals into segments and utilized the predicted probability of each segment. As a result, the counting algorithm we developed could report the frequency of wheezing during entire clinical lung sounds without any additional information, such as respiration volume.

Despite our contributions, several methodological limitations must be addressed. First, due to the limited availability of reference lung sound data collected in good quality, it is not sufficiently verified whether the developed algorithm would function well on the data collected from various patients in diverse recording environments. With useful methods such as assessment of lung sound quality, we would utilize the fine quality of lung sound data, resulting in a more sophisticated and accurate AI model. Furthermore, the algorithm must be enhanced and adjusted based on the clinical trials of long-term lung sound monitoring with a broad patient group in order to assure the validity, reliability, and applicability of preventive treatment in clinical and non-clinical settings. Second, this study does not provide empirical evidence on how sensitive the proposed algorithm is to different types of lung sounds. The frequency of wheezes was the focus of this study because it is known that wheezes might exacerbate asthma and COPD. A recently developed wearable stethoscope including a de-nosing function [53] could enable the widespread practical use of our counting algorithm when more adventitious sounds such as crackling, rhonchi, stridor, and pleural rub are accumulated by the device. In general, automated long-term monitoring via AI-based algorithms could assist preventative medicine by acquiring precursory information from numerous signals and images from the human body for the relevant bad health impacts. As so, long-term monitoring of wheezing occurrences and patterns may shed light on the development of various respiratory illness outcomes if combined with a patient's clinical records, such as symptom exacerbation and response to treatment. The integration of AI-based algorithms into long-term monitoring

could revolutionize preventative medicine. By acquiring precursory information from numerous signals and images from the human body, AI could potentially predict adverse health impacts. In the context of respiratory health, long-term monitoring of wheezing occurrences and patterns, combined with a patient's clinical records, could provide invaluable insights into the development of various respiratory illness outcomes. This could lead to more personalized treatment plans and improved patient outcomes.

## Conclusion

This research presents a deep learning-based algorithm for counting wheezes, utilizing a 1D-CNN-LSTM model. The model is trained on a variety of reference lung sound databases to predict the probability of abnormal sounds in each segment. Our algorithm then uses this model to count wheeze instances from recorded lung sounds and validates in real-time lung sound simulation.

Our wheeze counting method is straightforward yet effective, with potential for expansion into automatic symptom monitoring. This could be crucial in predicting the onset or severity of future abnormalities, as well as detecting current symptoms. Given the possible link between wheeze occurrence trends and symptom exacerbations, our approach could aid in preventing urgent emergencies like asthma attacks. Unlike traditional lung sound classification algorithms, our method can handle continuous data. With a detection accuracy of 90%, the results include identifying the number of total breath cycles and the proportion of abnormal sounds, along with real-time counting and visualization of these events throughout whole respiration. This could revolutionize research on predicting lung diseases based on long-term breathing patterns and offers utility in both clinical and non-clinical settings for immediate detection and remote intervention of worsened respiratory symptoms. Moreover, our counting algorithm can easily adapt to other bio-signals. For instance, when used with ECG (Electrocardiogram) or EMG (Electromyography) signals, it could automatically detect the intensity of heart or muscle anomaly patterns.

In conclusion, our study introduces a novel and effective approach to real-time wheeze detection and counting, which has significant potential for improving self-symptom management and telemedicine-based remote monitoring. This innovative wheeze counter, with its high detection accuracy and ability to handle continuous data, could play a crucial role in predicting lung diseases based on long-term breathing patterns. Furthermore, its adaptability to other bio-signals suggests a wide range of potential applications in both clinical and non-clinical settings. Future research should focus on further refining the algorithm and exploring its potential in various healthcare contexts.

## Supporting information

**S1 Table. Research trends in lung sound analysis and related papers.**
(DOCX)

**S2 Table. Model parameter values of the 1D CNN + LSTM model.**
(DOCX)

**S1 Fig. Comparison of wheeze between normal by raw signal and Mel spectrogram.** In some cases, there is coexistence of normal and wheeze sound in isolated breathing cycle.
(TIF)

**S2 Fig. Parametric study in number of Mel frequency bin, frame length, and labeling ratio.** Among 9 cases of parametric study, we choose parameters of case 4 to utilize in counting

algorithm.
(TIF)

**S3 Fig. Comparison of Sci-kit Learn classifiers between 1D CNN+LSTM model.** The classifier also trained by 10-fold cross-validation method.
(TIF)

**S4 Fig. Probability visualization of 3 comparative classifiers.** (A) Random Forest classifier, (B) K-Nearest Neighbors, (C) Multi-layer perceptron.
(TIF)

**S5 Fig. Test data overlapped with hospital noise (waiting room).** (A) original raw signal of test data, (B) test data with overlapped noise (SNR -20dB) is depicted in blue line, and result after noise reduce is plotted in orange line (The number of standard deviations above the noise is set to '0.1', and mode of stationary set to 'True'), and green line (default setting from library).
(TIF)

**S6 Fig. Probability visualization of noisy test data.** (A) prediction probabilities of original noisy data, (B) predictions of noise reduced data (The number of standard deviations above the noise is set to '0.1', and mode of stationary set to 'True'), and (C) different setting of noise reduced data (default setting from library).
(TIF)

**S1 Video. Real-time counting in simulation.**
(MP4)

## Author Contributions

**Conceptualization:** Sunghoon Im, Taewi Kim, Dohyeong Kim, Daeshik Kang, SungChul Seo.

**Data curation:** Sunghoon Im.

**Formal analysis:** Taewi Kim.

**Funding acquisition:** Daeshik Kang, SungChul Seo.

**Investigation:** Sunghoon Im, Taewi Kim, Yeonwook Roh, Changhwan Kim, Minho Kim, KyungMin Shim.

**Methodology:** Sunghoon Im, Taewi Kim, Yeonwook Roh, Changhwan Kim, Minho Kim, Jesung Koh, Seungyong Han, JaeWang Lee, Dohyeong Kim, Daeshik Kang, SungChul Seo.

**Project administration:** Daeshik Kang, SungChul Seo.

**Resources:** Sanghun Kang.

**Software:** Choongki Min, Sanghun Kang.

**Supervision:** Dohyeong Kim, Daeshik Kang, SungChul Seo.

**Validation:** Sunghoon Im, Seung Hyun Kim.

**Visualization:** Choongki Min.

**Writing – original draft:** Sunghoon Im.

**Writing – review & editing:** Sunghoon Im, Dohyeong Kim.

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
