## [Decision Letter · Decision Letter 0]

19 Jun 2023

PONE-D-23-13973Real-time counting of wheezing events from lung sounds using deep learning algorithms: implications for disease prediction and early interventionPLOS ONE

Dear Dr. Kang,

Thank you for submitting your manuscript to PLOS ONE. After careful consideration, we feel that it has merit but does not fully meet PLOS ONE’s publication criteria as it currently stands. Therefore, we invite you to submit a revised version of the manuscript that addresses the points raised during the review process.

We look forward to receiving your revised manuscript.

Kind regards,

Mohammad Amin Fraiwan

Academic Editor

PLOS ONE

Journal Requirements:

Reviewers' comments:

Reviewer's Responses to Questions

**Comments to the Author**

1. Is the manuscript technically sound, and do the data support the conclusions?

Reviewer #1: Partly

Reviewer #2: Partly

2. Has the statistical analysis been performed appropriately and rigorously? 

Reviewer #1: No

Reviewer #2: No

3. Have the authors made all data underlying the findings in their manuscript fully available?

Reviewer #1: No

Reviewer #2: Yes

4. Is the manuscript presented in an intelligible fashion and written in standard English?

Reviewer #1: Yes

Reviewer #2: Yes

5. Review Comments to the Author

Reviewer #1: Real-time counting of wheezing events from lung sounds using deep learning

algorithms: implications for disease prediction and early intervention

----------------

The paper proposes to use the 1d-cnn to count the wheeze events from lung sound dataset.

The problem of study is interesting and relevant.

However, the paper requires several issues to be fixed for the publication.

1. The graphics are poor, which is difficult to understand.

2. Authors are suggested discussing and citing some state of the art sound classification papers published in top journals.

https://ieeexplore.ieee.org/abstract/document/9784899

https://ieeexplore.ieee.org/abstract/document/9931407

https://ieeexplore.ieee.org/abstract/document/9684869

These papers explain the cnn-lstm, mfcc, ensemble learning for the different kinds of audio sound (eg. lung sound, etc.) classification, which are closely related to the

current study.

3. It is unclear how the current study is applicable unlike previously proposed model in the field.

4. The paper requires to perform the comparative study with the state of the art methods, which could establish trustworthiness.

5. The dataset is small, please use larger datasets.

6. The paper requires statistical significance test.

Reviewer #2: This is an interesting paper where authors propose a real-time wheezing sound counting algorithm based on a one-dimensional convolutional neural network and a long short-term memory combined (1D-CNN-LSTM) network model. In general, the paper is well-written and technically sound, and addresses a problem that may be of high-interest to the biomedical signal processing community. However, the opinion of this reviewer is that the following concerns need to be carefully resolved before resubmission.

Major comments:

1) First, in the Introduction, the authors present the state of the art in automatic detection and classification of adventitious lung sounds (second and third paragraphs of the section 1), but I think it is not complete, since some important references are missing. For example, in the last decade, algorithms based on non-negative matrix factorization (NMF) have been widely used in the detection-classification of wheezing sound.

2) I strongly believe that the database used by the authors is rather limited. Specifically, they have constructed a database consisting of 46 respiratory cycles (13 normal respiration/33 wheezing respiration). Then using data augmentation techniques, they have obtained a total of 5852 respiratory segments of 25 ms labelled as normal, wheeze or break. In this sense, my main concern is whether the dataset is sufficiently representative (has enough variability) to validate the performance of the proposed algorithm. In line 335 of the manuscript, the authors note the limited availability of reference lung sounds. However, in recent years “the publicly available ICBHI 2017 Challenge dataset [A]” has been widely used to assess the performance of adventitious sound detection/classification algorithms. I strongly encourage authors to use this database or a part of it to complete the database proposed by the authors.

[A] ICBHI 2017 challenge, respiratory sound database, 2017,

https://bhichallenge.med.auth.gr/ICBHI_2017_Challenge

3) On the other hand, the training and validation model proposed by the authors is often not used to evaluate the performance of algorithms due to its limitations. The authors simply divide the dataset into train and test. But, this validation methodology is highly dependent on the data used in both groups. That is, the performance obtained can vary significantly when the train and test group is composed of other data. In this sense, I recommend the authors to use a cross-validation methodology to reduce the variance of the estimated performance metrics. For example, a 10-fold cross-validation methodology.

4) Finally, the authors do not mention how the proposed algorithm performs in noisy environments. In my opinion, I think it would be quite interesting for the authors to show the performance obtained by the proposed algorithm when the analyzed lung sounds are overlapped with real clinical sounds. Authors can find clinical sounds recorded in real environments available online, for example: https://www.soundsnap.com/tags/clinic

Minor comments:

1) Could you add a link for downloading your approach?

2) Could you add a link for downloading the datasets used in your paper?

6. PLOS authors have the option to publish the peer review history of their article (what does this mean?). If published, this will include your full peer review and any attached files.

Reviewer #1: No

Reviewer #2: No

---

## [Author Response · Author response to Decision Letter 0]

11 Aug 2023

*Please read response letter (Response to Reviewers.docx) to find more detailed and sufficient responses.*

Response to Reviewer 1

General comments:

The paper proposes to use the 1d-cnn to count the wheeze events from lung sound dataset.

The problem of study is interesting and relevant.

However, the paper requires several issues to be fixed for the publication.

Our Response:

We appreciate you to accepting the definition of the issues. We spent plenty of time debating how to respond to your insightful comments, and we responded by highlighting each one with blue. Additionally, we highlighted our revised manuscript with red. We hope that our responses sufficiently satisfy your comments.

Comment 1:

The graphics are poor, which is difficult to understand.

Our response:

We would like to express our appreciation to the reviewer for providing intuitive feedback. We checked suggested papers in your comment (2) and find intuitive diagram of designed model. We rearrange the graphics to our Figure 3.

Comment 2:

Authors are suggested discussing and citing some state of the art sound classification papers published in top journals.

https://ieeexplore.ieee.org/abstract/document/9784899

https://ieeexplore.ieee.org/abstract/document/9931407

https://ieeexplore.ieee.org/abstract/document/9684869

These papers explain the cnn-lstm, mfcc, ensemble learning for the different kinds of audio sound (eg. lung sound, etc.) classification, which are closely related to the current study.

Our response:

We would like to express our appreciation to the reviewer for providing valuable feedback. As you mentioned, suggested papers are somewhat in the line with our research. Among the papers, two papers are about evaluating neonatal bowel sound, and it is very interesting that the bowel sound could be utilized as vital cue for infant care. Our research is, however, about lung sound monitoring by counting the frequency of wheezing evoked in COPD or Asthma patient, explored to environmental affect about respiratory diseases. Othe other hand, the other paper suggested is about real-time lung sound quality assessment for telehealth applications, so we decided to cite the paper of lung sound quality assessment in our discussion section where we discuss our research’s limitation and future work.

Comment 3:

It is unclear how the current study is applicable unlike previously proposed model in the field.

Our response:

We thank to your comment to explain our research in detail. Our paper suggests the counting algorithm of the wheeze sounds. We didn’t mean to suggest new model of classification, but the applicable method for practical using by our proposed algorithm. We use the predictions of segment-based AI model such as CNN-LSTM combined model which we used in our research and interpret the results of prediction to monitor the frequency and timing of wheeze sounds. We didn’t insist that our model is more suitable than other models, but we suggest the applicable methods to use AI models. It is important point to convince post-readers of our paper, so we added more clear explanation about what we did in the paper at discussion section.

Comment 4:

The paper requires to perform the comparative study with the state of the art methods, which could establish trustworthiness.

Our response:

We thank to your review, we additionally compared several models from ensemble, decision tree which are applicable to our counting algorithm. The results are summarized in Supplementary Figure 4.

Comment 5:

The dataset is small, please use larger datasets.

Our response:

Thanks for intuitive comment. As so, we additionally added ICBHI 2017 challenge datasets to convince our counting algorithm. Among the datasets, we used annotated wheeze signals.

Comment 6:

The paper requires statistical significance test.

Our response:

Thank you for your valuable comment, but our research doesn’t have any data that would make a statistical significance test required. Instead, we compared different AI models based on how well they classified segmented based prediction of the test data. Supplementary Figure 4 provides a summary of the results. (Please refer to the updates provided in the response to your comment 4.)

*Please read response letter (Response to Reviewers.docx) to find more detailed and sufficient responses.*

Response to Reviewer #2

General comments:

This is an interesting paper where authors propose a real-time wheezing sound counting algorithm based on a one-dimensional convolutional neural network and a long short-term memory combined (1D-CNN-LSTM) network model. In general, the paper is well-written and technically sound, and addresses a problem that may be of high-interest to the biomedical signal processing community. However, the opinion of this reviewer is that the following concerns need to be carefully resolved before resubmission.

We appreciate you to accepting the definition of the issues and suggest valuable comments. We spent plenty of time debating how to respond to your insightful and constructive comments. We responded by highlighting each one with blue. And additionally, we highlighted our revised manuscript with red. We hope that our responses sufficiently satisfy your comments.

Comment 1:

First, in the Introduction, the authors present the state of the art in automatic detection and classification of adventitious lung sounds (second and third paragraphs of the section 1), but I think it is not complete, since some important references are missing. For example, in the last decade, algorithms based on non-negative matrix factorization (NMF) have been widely used in the detection-classification of wheezing sound.

Our response:

We would like to express our appreciation to the reviewer for providing valuable feedback. As you mentioned, suggested method of algorithm NMF and related research is quite worth mentioning in introduction. So, we decided to cite paper about NMF in our introduction.

Comment 2:

I strongly believe that the database used by the authors is rather limited. Specifically, they have constructed a database consisting of 46 respiratory cycles (13 normal respiration/33 wheezing respiration). Then using data augmentation techniques, they have obtained a total of 5852 respiratory segments of 25 ms labelled as normal, wheeze or break. In this sense, my main concern is whether the dataset is sufficiently representative (has enough variability) to validate the performance of the proposed algorithm. In line 335 of the manuscript, the authors note the limited availability of reference lung sounds. However, in recent years “the publicly available ICBHI 2017 Challenge dataset [A]” has been widely used to assess the performance of adventitious sound detection/classification algorithms. I strongly encourage authors to use this database or a part of it to complete the database proposed by the authors.

[A] ICBHI 2017 challenge, respiratory sound database, 2017,

https://bhichallenge.med.auth.gr/ICBHI_2017_Challenge

Our response:

Thanks for intuitive comment. As so, we additionally added ICBHI 2017 challenge datasets to convince our counting algorithm. Among the datasets, we chose data from Asthma and COPD patients and healthy patients. As a result, we could train the model with 10 times bigger than prior datasets.

Comment 3:

On the other hand, the training and validation model proposed by the authors is often not used to evaluate the performance of algorithms due to its limitations. The authors simply divide the dataset into train and test. But, this validation methodology is highly dependent on the data used in both groups. That is, the performance obtained can vary significantly when the train and test group is composed of other data. In this sense, I recommend the authors to use a cross-validation methodology to reduce the variance of the estimated performance metrics. For example, a 10-fold cross-validation methodology.

Our response:

Thanks for intuitive and constructive feedback. I’d like to explain clearer about our evaluation process. As you suggested we trained the model with 10-fold cross-validation method. Then, among the trained model, we chose the model of highest accuracy. And for evaluation, we used totally un-seen test data, that is not used during the training process.

Comment 4:

Finally, the authors do not mention how the proposed algorithm performs in noisy environments. In my opinion, I think it would be quite interesting for the authors to show the performance obtained by the proposed algorithm when the analyzed lung sounds are overlapped with real clinical sounds. Authors can find clinical sounds recorded in real environments available online, for example: https://www.soundsnap.com/tags/clinic

Our response:

Thank you for your considerate review to upgrade our research. As your comment, we tested our algorithm when hospital noise is overlapped.

Comment 5, 6:

Could you add a link for downloading your approach? Could you add a link for downloading the datasets used in your paper?

Our response:

Of course, we will open our GitHub repository to approach our method and data after the paper is completely published (https://github.com/sunghoon-most/Wheeze_Counter).

---

## [Decision Letter · Decision Letter 1]

14 Sep 2023

PONE-D-23-13973R1Real-time counting of wheezing events from lung sounds using deep learning algorithms: implications for disease prediction and early interventionPLOS ONE

Dear Dr. Kang,

Thank you for submitting your manuscript to PLOS ONE. After careful consideration, we feel that it has merit but does not fully meet PLOS ONE’s publication criteria as it currently stands. Therefore, we invite you to submit a revised version of the manuscript that addresses the points raised during the review process.

ACADEMIC EDITOR: **The editor would like to commend the authors for not including the citation requests made by some of the reviewers from the first round. However, Kindly, remove references 1 and 70 in the revised version as the this does not meet the journal criteria. This reviewer has been excluded from the process.**

We look forward to receiving your revised manuscript.

Kind regards,

Mohammad Amin Fraiwan

Academic Editor

PLOS ONE

Reviewers' comments:

Reviewer's Responses to Questions

**Comments to the Author**

1. If the authors have adequately addressed your comments raised in a previous round of review and you feel that this manuscript is now acceptable for publication, you may indicate that here to bypass the “Comments to the Author” section, enter your conflict of interest statement in the “Confidential to Editor” section, and submit your "Accept" recommendation.

Reviewer #2: All comments have been addressed

Reviewer #3: All comments have been addressed

Reviewer #4: (No Response)

Reviewer #5: All comments have been addressed

2. Is the manuscript technically sound, and do the data support the conclusions?

Reviewer #2: Yes

Reviewer #3: Yes

Reviewer #4: No

Reviewer #5: Yes

3. Has the statistical analysis been performed appropriately and rigorously? 

Reviewer #2: Yes

Reviewer #3: Yes

Reviewer #4: No

Reviewer #5: Yes

4. Have the authors made all data underlying the findings in their manuscript fully available?

Reviewer #2: Yes

Reviewer #3: Yes

Reviewer #4: No

Reviewer #5: Yes

5. Is the manuscript presented in an intelligible fashion and written in standard English?

Reviewer #2: Yes

Reviewer #3: Yes

Reviewer #4: No

Reviewer #5: No

6. Review Comments to the Author

Reviewer #2: I am satisfied with the revision and I appreciate the work carried out by the authors to satisfy my comments. I recommend the paper for publication.

Reviewer #3: The addition of the ICBHI 2017 challenge datasets, adoption of the 10-fold cross-validation method, and testing the algorithm's robustness in noisy environments were especially appreciated.

Your commitment to sharing your approach and datasets with the community via your GitHub repository after the paper's publication is commendable. This openness aligns with our journal's principles of transparency and reproducibility in scientific research.

Reviewer #4: 1. Novelty, number of samples tested, datasets used, result metrics etc must be specified in the abstract

2. Sufficient novelty is to be established

3. Conclusion section could not be found.

4. What is the training and testing ratio? Datasplit ratio is done on what basis?

5. How do the authors mitigate overfitting?

6. How to validate the model?

7. Compariosn with similar methods would be beneficial

8. Implicaions from the survey, reserach gaps identified, how this work addresses those must be spelled clearly

9. Need content reorganization

10. Weak discussion

Reviewer #5: Please proof read the paper before it can be accepted.

Please recheck the format of the refences and citations.

7. PLOS authors have the option to publish the peer review history of their article (what does this mean?). If published, this will include your full peer review and any attached files.

Reviewer #2: No

Reviewer #3: No

Reviewer #4: No

Reviewer #5: No

---

## [Author Response · Author response to Decision Letter 1]

20 Oct 2023

*Please read response letter (Response to Reviewers.docx) to find more detailed and sufficient responses.*

Response to Reviewer 2

General comments:

I am satisfied with the revision, and I appreciate the work carried out by the authors to satisfy my comments. I recommend the paper for publication.

Our response: 

We appreciate your insightful comments, and we would like to express our gratitude to your time and attention in helping us to enhance the quality of our work.

Response to Reviewer 3

General comments:

The addition of the ICBHI 2017 challenge datasets, adoption of the 10-fold cross-validation method and testing the algorithm's robustness in noisy environments were especially appreciated.

Our response: 

We appreciate your insightful comments, and we would like to express our gratitude to your time and attention in helping us to enhance the quality of our work.

Response to Reviewer 4

General comments:

The paper proposes to use the 1d-cnn to count the wheeze events from lung sound dataset.

The problem of study is interesting and relevant.

However, the paper requires several issues to be fixed for the publication.

Our response:

We appreciate you to accepting the definition of the issues. We spent plenty of time debating how to respond to your insightful comments, and we responded by highlighting each one with blue. Additionally, we highlighted our revised manuscript with red. We hope that our responses sufficiently satisfy your comments.

Comment 1:

Novelty, number of samples tested, datasets used, result metrics etc must be specified in the abstract.

Our response:

We thank for your valuable comments. We completely modified the abstract according to the comments and specified novelty, number of samples tested, datasets used, result metrics etc.

Comment 2:

Sufficient novelty is to be established. 

Our response:

We appreciate your valuable comment. Our paper presents a cutting-edge segmentation algorithm that identifies and counts abnormal respiratory events, aiding in the diagnosis and monitoring of lung diseases. This innovative method, which uses a segment-based supervised learning approach and an AI-based classifier, is trained on three types of labeled lung sound data and focuses on long-term breathing patterns, enhancing its potential for early diagnosis and remote treatment of respiratory diseases.

Comment 3:

Conclusion section could not be found.

Our response:

Thank you for your valuable comment. We have incorporated some of the discussion and added content to the Conclusion section. This not only succinctly conveys the main points of the text but also includes how this research will impact other studies.

Comment 4:

What is the training and testing ratio? Data split ratio is done on what basis?

Our response:

We thankful for your valuable comment. In our model, we didn't follow the traditional train-test split ratio. Instead, we used the k-fold cross-validation method which divides the entire dataset into 10 equal parts or 'folds'. Each fold serves as the testing set once, and the remaining 9 folds serve as the training set. So, the data split ratio varies in each iteration of the training and testing process. This method ensures that every data point gets tested at least once and helps in minimizing bias. The use of 10-folds is not arbitrary but is a common practice in machine learning. It is a good balance between computational cost and model performance. If we increase the number of folds, the variance of the resulting estimate decreases, but the bias increases. Conversely, if we decrease the number of folds, the bias decreases, but the variance increases. Therefore, 10-folds is often chosen as it provides a good trade-off between bias and variance.

Comment 5:

How do the authors mitigate overfitting?

Our response:

Thank you for your comment. We used k-fold cross-validation method for our model validation, which does not follow the traditional train-test split ratio. Instead, it divides the entire dataset into 10 equal parts or 'folds'. 

In this method, the model is trained and tested 10 times, each time using a different fold as the testing set and the remaining 9 folds as the training set. This way, each data point gets to be in the testing set exactly once and in the training set 9 times. This process helps in minimizing the bias as we are using most of the data for fitting, and also, we are ensuring that every data point gets tested at least once. The advantage of using k-fold cross-validation is that all the entries in the original dataset are used for both training and testing. This provides a more accurate measure of how well the model generalizes to unseen data, which is the ultimate goal of a machine learning model. The choice of the number of folds depends on the size of the dataset. In our case, we chose 10 folds, which is a common choice in machine learning, as it offers a good trade-off between computational cost and model performance.

Comment 6:

How to validate the model?

Our response:

Model validation is a crucial step in the machine learning process to ensure that your model has learned effectively from the training data and can generalize well to new, unseen data. 

In our case, we employed the k-fold cross-validation technique for validating our model. This method works by dividing the entire dataset into 'k' equal segments or 'folds'. For example, if k equals 10, the dataset is split into 10 equal parts. The model is subsequently trained and tested 'k' times. During each cycle, one of the 'k' subsets is utilized as the test set, while the remaining 'k-1' subsets are combined to create a training set. The performance of the model is then averaged across the 'k' cycles to offer a more reliable measure of its effectiveness. This technique ensures that every data point in the dataset is used for both training and testing at least once, offering a thorough evaluation of the model's performance.

Comment 7:

Comparison with similar methods would be beneficial.

Our response:

Thank you for your kind comment. We have compiled the research trends in lung sound analysis and related papers into a single table. The table has been added to the supplementary materials.

Comment 8:

Implications from the survey, research gaps identified, how this work addresses those must be spelled clearly.

Our response:

Thank you for your positive comment. We provide specific explanations of the implications derived from the survey, the research gaps identified, and how this work addresses those issues. These details have been incorporated into the main text's introduction.

Implications:

The implications of this research are significant for the field of respiratory medicine. This study demonstrates the potential of AI-based technology in diagnosing and monitoring lung diseases. The development of a real-time event counting algorithm that can identify and classify abnormal breathing sounds, record their frequency and pattern over a specific period, and present this information in real-time, could revolutionize the way lung diseases are diagnosed and monitored. This could lead to earlier detection of lung diseases, more effective treatment plans, and improved patient outcomes.

Research Gaps:

Despite the advancements in AI-based lung sound analysis, there are still several research gaps that need to be addressed. Most existing models focus on the automatic diagnosis of a single recorded data, and applications to real-time monitoring data are still limited. They are developed based on short-term learning data and are not adaptable for real-time, continuous long-term signals. Additionally, most models focus on detecting the presence or absence of abnormalities at each respiratory unit, rather than analyzing the pattern and frequency of abnormal lung sounds over a longer period. 

Addressing the Gaps:

This study addresses these research gaps by developing a real-time event counting algorithm that can identify and classify abnormal breathing sounds, record their frequency and pattern over a specific period, and present this information in real-time. The algorithm is designed to be adaptable for real-time, continuous long-term signals, and focuses on analyzing the pattern and frequency of abnormal lung sounds over a longer period, rather than simply detecting the presence or absence of abnormalities at each respiratory unit. This represents a significant advancement in the field of AI-based lung sound analysis, and could have important implications for the diagnosis and monitoring of lung diseases.

Comment 9:

Need content reorganization.

Our response:

We appreciate your kind comment. We have moved the Model structure and Clinical lung sound data in this study from the Method section to the Results section and added a Conclusion and a Acknowledgements section.

Comment 10:

Weak discussion.

Our response:

We’re thankful for your valuable comment. We have supplemented the weak discussion by adding more content.

Response to Reviewer 5

General comments:

Please proofread the paper before it can be accepted. Please recheck the format of the refences and citations.

Our response: 

We appreciate your valuable comments. As your suggested, we rechecked our manuscript before resubmitting. We would like to express our gratitude to your time and attention in helping us to enhance the quality of our work.

---

## [Editor Report · Decision Letter 2]

2 Nov 2023

Real-time counting of wheezing events from lung sounds using deep learning algorithms: implications for disease prediction and early intervention

PONE-D-23-13973R2

Dear Dr. Kang,

We’re pleased to inform you that your manuscript has been judged scientifically suitable for publication and will be formally accepted for publication once it meets all outstanding technical requirements.

Kind regards,

Mohammad Amin Fraiwan

Academic Editor

PLOS ONE
---

## [Editor Report · Acceptance letter]

9 Nov 2023

PONE-D-23-13973R2 

Real-time counting of wheezing events from lung sounds using deep learning algorithms: implications for disease prediction and early intervention 

Dear Dr. Kang:

I'm pleased to inform you that your manuscript has been deemed suitable for publication in PLOS ONE. Congratulations! Your manuscript is now with our production department. 

Kind regards, 

on behalf of

Dr. Mohammad Amin Fraiwan 

Academic Editor

PLOS ONE